# Pre-Admission Antiplatelet Therapy in Cryptogenic Stroke: A Double-Edged Sword

**DOI:** 10.3390/jcm14041061

**Published:** 2025-02-07

**Authors:** Jessica Seetge, Balázs Cséke, Zsófia Nozomi Karádi, Edit Bosnyák, László Szapáry

**Affiliations:** 1Stroke Unit, Department of Neurology, University of Pécs, 7624 Pécs, Hungary; j.seetge@gmx.de (J.S.); karadi.zsofia@pte.hu (Z.N.K.); bosnyak.edit@pte.hu (E.B.); 2Department of Emergency Medicine, University of Pécs, 7624 Pécs, Hungary; cseke.balazs@pte.hu

**Keywords:** pre-admission antiplatelet therapy, cryptogenic stroke, 90-day mRS

## Abstract

**Background**: Cryptogenic stroke, a challenging subtype of acute ischemic stroke (AIS), is characterized by the absence of an identifiable etiology despite thorough diagnostic assessment. The role of pre-admission antiplatelet therapy (APT) in this population remains poorly understood, as current guidelines are primarily based on evidence from other stroke subtypes. Therefore, this study investigates the impact of pre-admission APT on functional outcomes in patients with cryptogenic stroke. **Methods:** A total of 224 patients with cryptogenic stroke admitted to the University of Pécs between February 2023 and September 2024 were retrospectively analyzed. Propensity score matching (PSM) with sensitivity analysis was employed to balance baseline characteristics, resulting in a matched cohort of 122 patients. Logistic regression and mediation analysis were used to evaluate the association between pre-admission APT and favorable outcome at 90 days, defined as a modified Rankin Scale (mRS) score of 0–2. **Results:** A favorable outcome was achieved by 39.3% of patients with pre-admission APT (*n* = 61), compared to 61.7% of those not receiving pre-admission APT (*n* = 162) (odds ratio [OR] = 0.40, 95% confidence interval [CI]: 0.22–0.74, *p* = 0.004). After PSM and adjusting for confounders, including pre-morbidity mRS (pre-mRS) (OR = 0.17, CI: 0.06–0.49, *p* < 0.001), National Institutes of Health Stroke Scale (NIHSS) at 72 h post-stroke (OR = 0.67, CI: 0.50–0.88, *p* = 0.004), and smoking status (OR = 0.14, CI: 0.02–0.78, *p* = 0.025), pre-admission APT remained associated with poorer functional outcomes (adjusted OR [aOR] = 0.21, 95% CI: 0.06–0.76, *p* = 0.018). **Conclusions:** Pre-admission APT is independently associated with poorer functional outcomes in cryptogenic stroke patients. These findings challenge traditional assumptions regarding APT’s protective role and highlight the need for prospective studies to refine its use in cryptogenic stroke management.

## 1. Introduction

Cryptogenic strokes, accounting for approximately 25–40% of acute ischemic strokes (AIS) [1,2,3], present a significant challenge in modern neurology. Unlike other ischemic stroke subtypes with well-defined causes, cryptogenic strokes are diagnosed in the absence of an identifiable etiology despite extensive diagnostic evaluation [4]. This diagnostic uncertainty complicates secondary prevention strategies, which therefore often rely on generalized guidelines rather than targeted, etiology-specific interventions [5].

Pre-admission antiplatelet therapy (APT), defined as antiplatelet medications prescribed and consistently taken by patients prior to hospital arrival, introduces additional complexity to this already challenging scenario. While APT remains the standard prevention method to reduce thrombotic risk in individuals with prior non-cardioembolic ischemic events [5,6], its impact on functional outcomes in patients who experience a cryptogenic stroke remains poorly understood. Studies comparing APT to anticoagulation therapy in patients with cryptogenic stroke, including embolic stroke of undetermined source (ESUS), have found both approaches to be similarly ineffective in preventing recurrent strokes, calling into question the overall efficacy of APT in this population [7,8]. Moreover, cryptogenic stroke patients frequently lack the traditional risk profiles observed in other stroke subtypes [9,10], raising concerns that pre-admission APT may inadvertently contribute to poorer functional outcomes.

This study investigates the association between pre-admission APT and 90-day functional outcomes in cryptogenic stroke patients. By examining this relationship, the findings aim to shed light on the complex role of pre-admission APT in this population and address critical gaps in understanding its impact on post-stroke recovery. These insights could contribute to the development of more targeted clinical guidelines for managing cryptogenic stroke patients with prior APT use.

## 2. Materials and Methods

### 2.1. Study Design and Patient Population

This retrospective study was conducted using data from the prospective Transzlációs Idegtudományi Nemzeti Laboratórium (TINL) STROKE registry. The study included 236 patients with cryptogenic stroke who were admitted to the Department of Neurology, University of Pécs, between February 2023 and September 2024. Cryptogenic stroke was defined as “an imaging-confirmed stroke with unknown source despite thorough diagnostic assessment” [4,5]. Patients were included in the absence of evidence for cardioembolism, large artery atherosclerosis (stenosis > 50%), small vessel disease, or atrial fibrillation detected on a 12-lead echocardiogram (ECG) or after 24 h cardiac monitoring. Additionally, patients who died before completing the diagnostic work-up (*n* = 12) were excluded to ensure that a full standard evaluation could be conducted, resulting in a final cohort of 224 patients.

The study protocol was reviewed and approved by the Scientific and Research Ethics Committee of the Medical Research Council of the University of Pécs (RRF-2.3.1-21-2022-00011, approval date: 1 September 2022) and the Scientific and Research Ethics Committee of the Medical Research Council of Hungary (BM/22444-1/2024, approval date: 1 September 2024) to ensure that it met all regulatory requirements and ethical guidelines, including participant privacy and data protection standards. All study procedures were carried out in compliance with applicable ethical guidelines, and ongoing monitoring by the ethics committees ensured adherence to approved protocols.

### 2.2. Data Collection and Measurements

Baseline characteristics included demographic data (age and sex) and clinical variables, such as pre-morbidity modified Rankin Scale (pre-mRS) scores and stroke severity, assessed using the National Institutes of Health Stroke Scale (NIHSS) at admission and 72 h post-stroke. Additionally, recorded factors included the classification of cryptogenic stroke subtypes (e.g., ESUS), onset-to-door times, and admission plasma glucose levels. Vascular risk factors (current smoking, alcohol use, and prior stroke history) and comorbidities, such as hypertension and diabetes mellitus, were also documented. Furthermore, treatment modalities for cryptogenic stroke (thrombolysis [TL], mechanical thrombectomy [MT], or combined therapy [TL + MT]) were recorded.

### 2.3. Outcome Measures

The primary endpoint of the study was a favorable functional outcome at 90 days, defined as a modified Rankin Scale (mRS) score of 0–2. Outcomes were evaluated through telephone interviews conducted by a physician or a certified neurology nurse 90 days post-admission.

### 2.4. Statistical Analyses

Data analysis was performed using Python (version 3.13.0, Python Software Foundation, Wilmington, DE, USA). Baseline characteristics between the groups were compared using Fisher’s exact test or $\chi^{2}$ test for categorical variables (reported as fractions and percentages) and independent sample t-tests, Analysis of Variance (ANOVA), or Kruskal–Wallis tests for continuous variables (reported as means ± standard deviation [SD] or medians with interquartile ranges [IQR]).

Propensity score matching (PSM) was performed using a nearest-neighbor approach to minimize baseline differences between patients receiving pre-admission APT (*n* = 61) and those not receiving pre-admission APT (*n* = 162). A caliper width of 0.1 was applied to ensure rigorous matching of patients with highly similar propensity scores, thereby reducing residual confounding. Following matching, 122 patients were included in the final cohort for outcome analysis.

Logistic regression was used to estimate the adjusted odds of achieving a favorable functional outcome, with results reported as odds ratios (OR) and 95% confidence intervals (CI). Mediation analysis was employed to evaluate the potential indirect effects of prior stroke history as a mediator between pre-admission APT and functional outcomes. The analysis decomposed the total effect into direct and indirect effects. For all statistical analyses, significance was defined as a *p*-value < 0.05.

## 3. Results

### 3.1. Baseline Characteristics

Before matching, patients with pre-admission APT (*n* = 61; aspirin: *n* = 29, clopidogrel: *n* = 27, dual antiplatelet therapy [DAPT]: *n* = 5) were older (mean age 69.87 ± 11.20 years vs. 64.76 ± 13.26 years, *p* = 0.005) and more likely to have a history of prior stroke (29.5% vs. 2.5%, *p* < 0.001), hypertension (86.9% vs. 72.8%, *p* = 0.042), and diabetes mellitus (41.0% vs. 21.0%, *p* = 0.004) than patients without pre-admission APT (*n* = 162) (Table 1).

### 3.2. Propensity Score Matching and Sensitivity Analysis

After PSM, 122 patients (*n* = 61 with pre-admission APT and *n* = 61 without pre-admission APT) were included in the final cohort. As shown in Table 1 and Figure 1, baseline characteristics were well balanced between the groups, with standardized mean differences (SMD) for all covariates below 0.1, indicating minimal residual confounding. Sensitivity analyses using stricter propensity score calipers (0.1) confirmed the robustness of the findings, with results consistent with the primary analysis.

### 3.3. Functional Outcomes

Before matching, the unadjusted analysis revealed that pre-admission APT was significantly associated with lower odds of achieving a favorable functional outcome (OR = 0.40, 95% CI: 0.22–0.74, *p* = 0.004): Only 39.3% of patients receiving pre-admission APT achieved a favorable outcome, compared to 61.7% of patients without pre-admission APT (Figure 2).

### 3.4. Logistic Regression and Mediation Analysis

As shown in Table 2, significant predictors of favorable outcome included lower pre-mRS score (OR = 0.17, 95% CI: 0.06–0.49, *p* < 0.001), lower NIHSS score at 72 h post-stroke (OR = 0.67, 95% CI: 0.50–0.88, *p* = 0.004), and non-smoking status (OR = 0.14, 95% CI: 0.02–0.78, *p* = 0.025).

Notably, even after adjusting for confounders, pre-admission APT remained independently associated with unfavorable functional outcomes at 90 days (adjusted odds ratio [aOR] = 0.21, 95% CI: 0.06–0.76, *p* = 0.018), indicating that patients receiving pre-admission APT had a 79% lower chance of achieving a favorable functional outcome compared to those without pre-admission APT (Figure 3).

Mediation analysis indicated that a prior history of stroke was not a significant mediator of the relationship between pre-admission APT and unfavorable functional outcome (*p* = 0.684). Both the average direct effect (ADE) and average causal mediation effect (ACME) estimates are nonsignificant, as shown in Table 3. These results suggest that the association between pre-admission APT and unfavorable outcome is predominantly direct rather than mediated by prior stroke history.

## 4. Discussion

This study demonstrates that pre-admission APT is independently associated with poorer 90-day functional outcomes in cryptogenic stroke patients, raising important questions about the underlying mechanisms and their implications for clinical management.

### 4.1. Comparison to the Current Literature

The existing literature indicates that approximately 50–60% of cryptogenic stroke patients achieve favorable long-term functional outcomes (mRS ≤ 2) [11,12,13,14], with lower or similar rates of functional dependency (mRS > 2) compared to non-cardioembolic stroke patients at discharge or 6 months, respectively [9,15].

However, the role of pre-admission APT in these outcomes remains uncertain. Research in general stroke populations has yielded mixed results: while some studies report no significant influence on stroke severity, in-hospital mortality, or long-term recovery [16,17,18,19,20], others suggest modest functional benefits at discharge [21,22,23] and improved 3-month mRS scores in selected patient groups [24]. In contrast, our findings align with studies in patients undergoing MT, where pre-admission APT has been associated with poorer functional outcomes (OR = 2.36, 95% CI: 1.03–5.54, *p* = 0.04) [25].

### 4.2. Possible Explanations

The mechanisms by which APT influences ischemic stroke outcomes extend beyond platelet aggregation. While platelets are central to thrombus formation, they also play critical roles in inflammation, immune responses, tissue repair, and the maintenance of vascular integrity [26]. Accordingly, platelet inhibitors provide protective effects not only by preventing clot formation, but also by stabilizing atherosclerotic plaque, promoting vascular dilation, and reducing oxidative stress and thrombus propagation [27].

However, these protective effects can act as a double-edged sword, potentially contributing to adverse outcomes in certain patients. In individuals with undiagnosed cerebrovascular abnormalities or subclinical vascular damage, often undetected during routine evaluations, APT may exacerbate microvascular fragility, increasing the risk of microbleeds and hemorrhagic transformation [28]. This vulnerability may be further aggravated by systemic conditions such as hypertension and diabetes mellitus, increasing the risk of adverse outcomes and impeding functional recovery [29].

While these proposed mechanisms are well supported by the existing literature, they remain theoretical within the context of our study, which was not specifically designed to directly investigate underlying biological pathways. However, the established evidence supporting these mechanisms provides a strong rationale for their consideration when interpreting our findings.

### 4.3. Clinical Implications

These findings underline the need to critically reassess APT use in cryptogenic stroke patients. Current stroke prevention guidelines, primarily developed for broader ischemic stroke populations, may not fully address the unique complexities of cryptogenic stroke. Given the potential for both protective and detrimental effects, clinicians must carefully balance the risks and benefits of APT in this population.

Future research should focus on prospective trials to clarify the role of pre-admission APT in cryptogenic stroke and identify subgroups that might benefit from its use. Mechanistic studies are also needed to explore how APT interacts with vascular pathology and systemic risk factors, influencing post-stroke recovery.

Incorporating advanced diagnostic tools into routine care could further refine treatment strategies. For example, the use of magnetic resonance imaging (MRI) to detect microbleeds or subtle vascular abnormalities, alongside biomarkers of vascular damage and extended cardiac monitoring to identify subclinical embolic sources, could improve risk stratification. These advancements would enable the development of personalized therapeutic strategies that optimize functional outcomes while minimizing unnecessary risks.

### 4.4. Limitations

This study has several limitations that should be considered when interpreting the results. The retrospective design and modest sample size may limit the generalizability of our findings. While PSM was used to minimize observed confounding, the potential influence of unmeasured confounders cannot be entirely excluded. In particular, while medication data were collected through pharmacy records and patient or caregiver reports, actual medication adherence could not be directly verified. Furthermore, although we adjusted for key comorbidities such as prior stroke, hypertension, and diabetes mellitus, the presence of undiagnosed conditions may still introduce residual confounding. Lastly, while standard diagnostic imaging was used to guide treatment decisions, advanced imaging metrics (e.g., Alberta Stroke Program Early CT Score [ASPECTS], modified CT Angiography Score [mCTA], and CT perfusion imaging [CTP] parameters) were not uniformly incorporated and may have provided further insights into outcome variability.

The reliance on the mRS as the primary outcome measure, while widely accepted, may not fully capture the multidimensional aspects of functional recovery. Dichotomizing outcomes into ‘favorable’ and ‘unfavorable’ categories, while practical for analysis, risks oversimplifying the complexity of post-stroke recovery. Future studies should consider continuous measures or multidimensional scales that incorporate cognitive, emotional, and physical recovery to provide a more comprehensive assessment of outcomes.

Additionally, the 90-day follow-up period may not fully reflect the long-term recovery trajectory. Longer follow-up periods could reveal whether patients not receiving pre-admission APT eventually achieve comparable outcomes to those on APT, potentially indicating an accelerated progression of unfavorable outcomes in the latter group.

## 5. Conclusions

Pre-admission APT is independently associated with poorer 90-day functional outcomes in cryptogenic stroke patients. Future prospective studies are necessary to reevaluate the role of APT in this population and to investigate the underlying mechanisms driving this association.

## Figures and Tables

**Figure 1 jcm-14-01061-f001:**
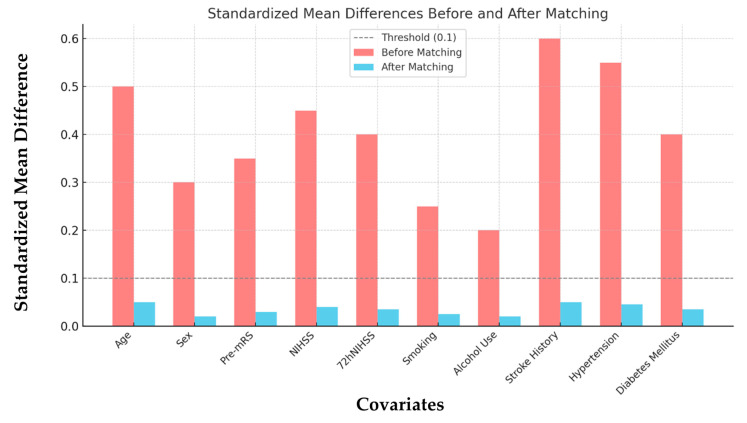
Standardized mean differences before and after propensity score matching. Abbreviations: Pre-mRS = pre-morbidity modified Rankin Scale, NIHSS = National Institute of Health Stroke Scale score at admission, 72hNIHSS = National Institute of Health Stroke Scale score 72 h post-stroke.

**Figure 2 jcm-14-01061-f002:**
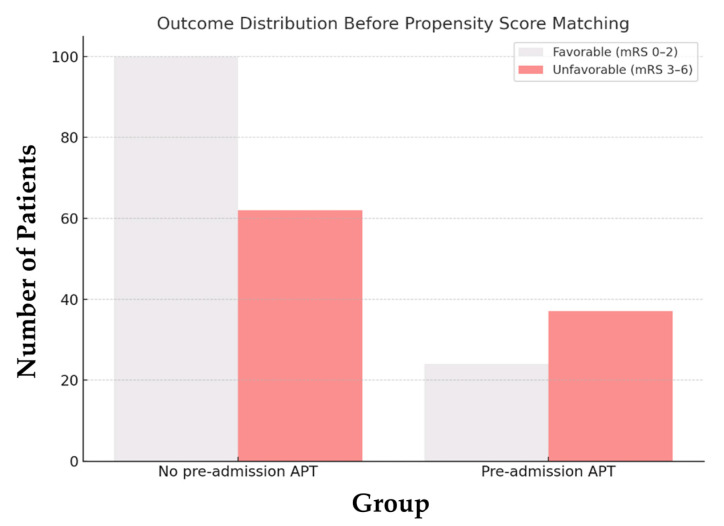
Distribution of favorable outcomes before propensity score matching. Abbreviations: mRS = modified Rankin Scale, APT = antiplatelet therapy.

**Figure 3 jcm-14-01061-f003:**
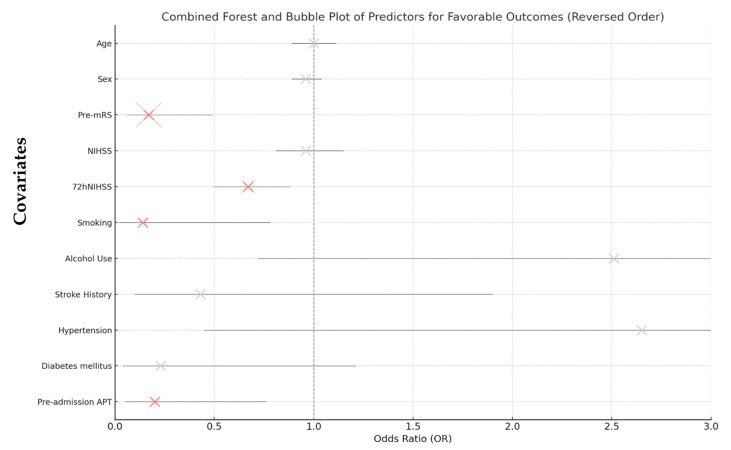
Combined forest and bubble plot of predictors of favorable outcome. Abbreviations: Pre-mRS = pre-morbidity modified Rankin Scale, NIHSS = National Institute of Health Stroke Scale score at admission, 72hNIHSS = National Institute of Health Stroke Scale score 72 h post-stroke, APT = antiplatelet therapy.

**Table 1 jcm-14-01061-t001:** Baseline characteristics before and after propensity score matching.

	Before Propensity Score Matching			After Propensity Score Matching		
	Pre-Admission APT *n* = 61	No Pre-Admission APT *n* = 162	*p*-Value	Pre-Admission APT *n* = 61	No Pre-Admission APT *n* = 61	*p*-Value
Patient Demographics						
Age, years, mean ± SD Sex, male, *n* (%)	69.87 ± 11.20 30 (49.2%)	64.76 ± 13.26 73 (45.1%)	0.005 0.690	69.87 ± 11.20 30 (49.2%)	72.10 ± 9.35 24 (39.3%)	0.235 0.362
Clinical Variables						
Pre-mRS score, mean ± SD NIHSS score, mean ± SD 72hNIHSS score, mean ± SD Etiology, ESUS, *n* (%) Onset-to-door time, median [IQR] Plasma glucose, mean ± SD	0.61 ± 1.24 6.5 ± 6.1 4.2 ± 5.2 11 (18.0%) 198 [93.0–390.0] 7.62 ± 2.47	0.46 ± 1.04 6.6 ± 5.5 4.1 ± 5.2 28 (17.3%) 242.0 [100.5–713.5] 7.12 ± 2.27	0.404 0.926 0.852 1.00 0.202 0.175	0.61 ± 1.24 6.5 ± 6.1 4.2 ± 5.2 11 (18.0%) 198 [93.0–390.0] 7.62 ± 2.47	1.05 ± 1.48 7.6 ± 5.7 4.7 ± 5.5 6 (9.8%) 150 [94.0–732.0] 7.66 ± 1.88	0.078 0.290 0.601 0.296 0.636 0.925
Medical History, *n* (%)						
Current smoking Alcohol use Stroke history Hypertension Diabetes mellitus	14 (23.0%) 24 (39.3%) 18 (29.5%) 53 (86.9%) 25 (41.0%)	57 (35.2%) 73 (45.1%) 4 (2.5%) 118 (72.8%) 34 (21.0%)	0.113 0.538 <0.001 0.042 0.004	14 (23.0%) 24 (39.3%) 18 (29.5%) 53 (86.9%) 25 (41.0%)	16 (26.2%) 20 (32.8%) 15 (24.6%) 56 (91.8%) 32 (52.5%)	0.833 0.572 0.684 0.557 0.276
Recanalization therapy, *n* (%)						
TL MT TL + MT	25 (41.0%) 10 (16.4%) 5 (8.2%)	48 (29.6%) 36 (22.2%) 11 (6.8%)	0.147 0.439 0.943	25 (41.0%) 10 (16.4%) 5 (8.2%)	31 (50.8%) 5 (8.2%) 4 (6.6%)	0.364 0.270 1.00

Abbreviations: APT = antiplatelet therapy, Pre-mRS = pre-morbidity modified Rankin Scale, NIHSS = National Institute of Health Stroke Scale score at admission, 72hNIHSS = National Institute of Health Stroke Scale score 72 h post-stroke, ESUS = embolic stroke of undetermined source, IQR = interquartile range, TL = thrombolysis, MT = mechanical thrombectomy.

**Table 2 jcm-14-01061-t002:** Predictors of favorable outcome after propensity score matching.

	OR	95% CI	*p*-Value
Patient Demographics			
Age, years Sex, male	0.96 1.03	0.89 to 1.04 0.30 to 3.55	0.308 0.964
Clinical Variables			
Pre-mRS score NIHSS score 72hNIHSS score Etiology, ESUS Onset-to-door time Plasma glucose	0.17 0.96 0.67 0.39 1.00 1.12	0.06 to 0.49 0.81 to 1.15 0.50 to 0.88 0.06 to 2.38 1.00 to 1.00 0.78 to 1.61	<0.001 0.674 0.004 0.306 0.663 0.536
Medical History			
Current smoking Alcohol use Stroke history Hypertension Diabetes mellitus	0.14 2.51 0.43 2.65 0.23	0.02 to 0.78 0.72 to 8.70 0.10 to 1.90 0.45 to 15.6 0.04 to 1.21	0.025 0.148 0.263 0.282 0.083
Recanalization therapy			
TL MT TL + MT	2.28 1.08 5.08	0.49 to 10.5 0.12 to 9.95 0.53 to 48.8	0.290 0.944 0.159

Abbreviations: OR = odds ratio, CI = confidence interval, pre-mRS = pre-morbidity modified Rankin Scale, NIHSS = National Institute of Health Stroke Scale score at admission, 72hNIHSS = National Institute of Health Stroke Scale score 72 h post-stroke, ESUS = embolic stroke of undetermined source, TL = thrombolysis, MT = mechanical thrombectomy.

**Table 3 jcm-14-01061-t003:** Mediation analysis results of prior stroke.

	Estimate	Lower CI	Upper CI	*p*-Value
ADE no pre-admission APT	−0.037	−0.210	0.128	0.684
ADE pre-admission APT	−0.037	−0.207	0.127	0.684
ACME no pre-admission APT	−0.015	−0.080	0.041	0.596
ACME pre-admission APT	−0.014	−0.080	0.039	0.596
Total effect	−0.051	−0.230	0.127	0.534

Abbreviations: CI = confidence interval, ADE = average direct effect, APT = antiplatelet therapy, ACME = average casual mediation effect.

## Data Availability

The original contributions presented in the study are included in the article and further inquiries can be directed to the corresponding author.

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
