# Peer review of "Pre-Admission Antiplatelet Therapy in Cryptogenic Stroke: A Double-Edged Sword"

_jcm, 2025, doi:10.3390/jcm14041061_

Round 1
Reviewer 1 Report
Comments and Suggestions for Authors
File attached

It can be improved.
Reviewer 2 Report
Comments and Suggestions for Authors
This study provides important insights into the impact of APT on the prognosis of patients with cryptogenic stroke. The results indicate that, contrary to traditional assumptions about the protective role of APT, pre-admission APT is associated with poorer functional outcomes within 90 days. The study employs propensity score matching and sensitivity analysis to adjust for confounding factors, including pre-stroke functional status, stroke severity, and smoking status. However, there are still some issues in the article that need to be addressed.
1. I suggest providing a clearer definition of "Pre-admission Antiplatelet Therapy," such as specifying it as the use of antiplatelet medications prescribed and continuously taken by patients before arriving at the hospital (e.g., at home or during outpatient follow-ups). The definition of the admission time point may vary across different countries and regions.
2. It would be better to include the types of antiplatelet medications used in the table for a more detailed presentation.
3. In Table 1, the standard deviation of Onset-to-door time greatly exceeds its mean, indicating a likely skewed distribution. The data should be presented as the median with interquartile range (IQR) and analyzed using non-parametric tests.
4. In Table 3, the estimates for ADE no pre-admission APT and ADE pre-admission APT are lower than the lower bound of their confidence intervals. Please double-check the accuracy of these data.
Round 2
Reviewer 2 Report
Comments and Suggestions for Authors
All the previous issues have been resolved, and I have no more questions.